# Numerical simulation of a hot-air cleaning fan for the combine harvester

**Tao Zhang** [1¶]*, **Guoliang You** [2¶], **Yaoming Li** [3]

1 School of Intelligent Equipment and Technology, Wuxi Taihu University, Wuxi, China, 2 School of Mechanical Engineering, Guizhou University, Guiyang, China, 3 School of Agricultural Engineering, Jiangsu University, Zhenjiang, China

¶ These authors contributed equally to this work.
* jsdxzt@foxmail.com

## Abstract

This study aimed to improve the separation efficiency of damp crops by developing a hot-air cleaning fan. First, the aerodynamic performance of two fans with different inlet structures was compared. The results showed a direct correlation between the static pressure and the outlet air velocity. The modified-inlet fan has an outlet average air velocity very close to the original fan of the combine harvester. Hence, the modified- inlet fan model was chosen for further analysis. Subsequent evaluations were conducted to assess the effects of fan speed, inlet air temperature, and volute temperature on fan performance. Fan speed exhibited a negative correlation with outlet air temperature but a positive correlation with outlet air velocity, at both the upper and lower outlets. Furthermore, both the inlet air temperature and the volute temperature were directly related to the air temperature at these outlets. A Response Surface Methodology (RSM) with three factors at three levels was utilized to optimize fan parameters, identifying fan speed and inlet air temperature as the predominant factors affecting outlet air velocity and temperature, respectively.The optimal set of parameters was determined to be a fan speed of 1445 rpm, an inlet temperature of 85 °C, and a volute temperature of 60 °C. Under these conditions, bench test results indicated the average air velocity and temperature are basically consistent with the theoretical optimal value.

## 1. Introduction

Combine harvesters are essential for modern agriculture as it can significantly enhance the efficiency of the grain harvesting process [1–2]. The harvesters' operational efficiency are directly impacted by the cleaning system. At present, the air-and-screen cleaning device is widely used in the cleaning system of combine harvesters [3], but due to the use of unheated airflow, the problem of screen blockage often occurs when cleaning crops with high moisture content.After the screen is completely clogged, the cleaning quality significantly deteriorates.

**Data availability statement:** All raw data are within the supplementary information.

**Funding:** This research was funded by National Natural Science Foundation of China under Grant grant number (51975257).

**Competing interests:** We ensured that there was no conflict of interest.

Numerous experts and scholars have conducted extensive studies on cleaning devices. Cui et al. [4] developed a double-fan air sieve-type cleaning apparatus. This design incorporated a three-outlet structure for the frontal fan, with a subsequent fan positioned at the trash churn's forefront, facilitating the discharge of short stalks at the sieve's end. Sanderson et al. [5]discovered a step cleaning mechanism that uses the shear force of vortices in the slot to aggregate particles on the surface of the cleaning medium to the inner corner of the slot to avoid blocking. Lawinska and Modrzewski [6] conducted a research in this regard and found that for the screen blocking problem caused by viscous materials, high-speed airflow can be used to remove stuck materials from the screen surface, achieving the goal of cleaning the screen. Yi.He, Andrew E. Bayly, and Ali Hassanpour [7] employed dynamic meshing with conventional CFD-DEM. The model tracks the movements of microscopic particles using CFD-DEM and uses dynamic meshing to capture flows produced by large mobile objects. It exhibits effective management of intricate free-moving boundaries and dynamic mesh partitioning. High agreement with experimental data from pertinent literature was obtained by the study through experimental validation of two classic gas-solid two-phase flow simulation cases. Mahmoud A El-Emam [8] used a combined discrete element method (DEM) and computational fluid dynamics (CFD) methodology in 2023 to examine the efficiency of cyclonic spiral inlets in airflow patterns and bioparticle separation processes. Variations in pressure, velocity, and turbulence parameters inside the swirling airflow demonstrated the study's conclusion that inlet type and design play a crucial influence in cyclonic separator performance.

In addition, the performance of the fan has a significant impact on the cleaning quality. Li [9] determine the structure of a single-outlet single-channel centrifugal fan, and numerically simulate its internal flow field to obtain the distribution of the fan's internal speed and pressure and other parameters, and to improve its internal flow field by changing the fan parameters.Pakari and Ghani [10] analysed variations in air velocity and volume flow in different ventilation systems using CFD. Xu and Li [11] employed CFD simulations to analyse the effects of fan guide plate angles, sieve openings, and threshing cylinder rotational speed on the airflow field within the cleaning shoe. Similarly, Chai [12] simulated the air distribution inside the fan and the airflow speed distribution at the fan outlet using CFD. Liang et al. [13] also applied CFD simulations to study airflow distribution within the cleaning shoe, validating their findings through hot-wire anemometry.

Zhang [14] have demonstrated the applicability of the hot airflow cleaning approach in improving the cleaning quality.This study aims to design and optimize a hot-air cleaning fan for combine harvesters. The core value lies in significantly boosting the cleaning performance by raising the fan intake temperature, ultimately helping farmers achieve higher yields.

## 2. Design of hot airflow fans

### 2.1. Structure and main parameters of the Hot Airflow Fan

The hot airflow cleaning fan designed in this paper is mainly composed of a centrifugal fan, two heating units on the side, a electric heating plate on the volute, as shown in Fig 1.

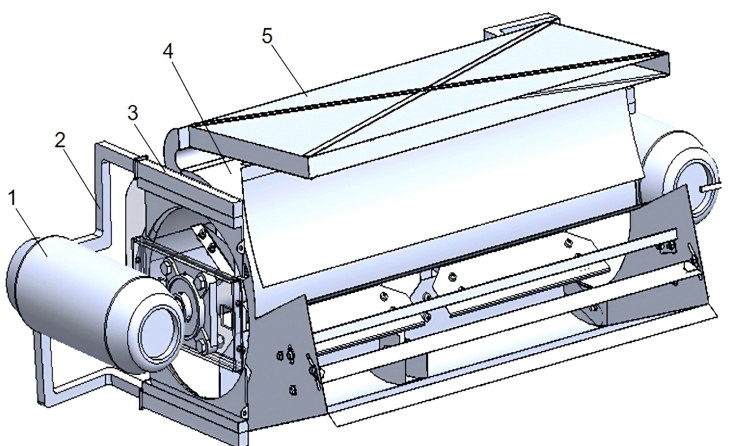

1.heater  2.Air intake duct  3.Fan baffle  4.Electric heating plate  5.centrifugal fan

**Fig 1. Structure of hot airflow fan.**

The volumetric flow rate of the cleaning airflow is primarily governed by the quantity of impurities to be eliminated from the material undergoing cleaning, as described by the following relationship [15]:

$$V = \frac{Q_m \beta}{\rho \mu} \tag{1}$$

where: V is the represents the volumetric airflow rate (m³/s); $Q_m$ is the denotes the machine feeding rate (kg/s); $\rho$ is the stands for the air density, with a typical value of 1.20 kg/m³; $\mu$ is the mixing concentration ratio of the airstream containing impurities, typically ranging between 0.2 and 0.3; $\beta$ is the straw ratio coefficient of the threshed rice outputs. When the machine is fully fed, $\beta$ typically ranges between 0.15 and 0.2 for crops, while for wheat and rice, the values generally lie between 0.1 and 0.15 [15]. From Equation (1), the airflow is determined to be 2.5 m³/s. The impeller width, denoted as b, is expressed in terms of $b_0$. b=(1.15 2.15) $b_0$. Here,$b_0$ represents the airflow width at the impeller inlet and can be computed as given in reference [16].

$$b_0 = \frac{D_0 \left(1 - K_d^2\right) \mu_0}{4D_0} = K_0 \sqrt[3]{\frac{V}{n - (1 - \varphi)\, \mu_0}} \tag{2}$$

From equation (2), the value of *b* is determined to be 0.127 m.

According to the fan design manual titled "Handbook of Agricultural Machinery Design (Upper Volume)" [17], the inlet diameter, $D_0$, has a specific relationship with the impeller's outer diameter, $D_2$, and its inner diameter, $D_1$. This relationship between $D_2$ and $D_1$ is as follows:

$$D_2 = \frac{D_0}{(0.65 \sim 0.8)} \tag{3}$$

$$D_1 = (0.5 \sim 0.6)D_2 \tag{4}$$

The Formula (3) and (4) gives $D_2 = 0.556$ m and $D_1 = 0.278$ m.

The overall dynamic pressure of the fan is influenced by the number of blades. Increasing the blade count can augment the air delivery capacity. However, for this study, taking into account the required air volume and the typical blade count of 4–6 for general-purpose centrifugal fans in agricultural machinery, a design with four blades was chosen, as illustrated in Fig 2a. The volute design aligns with the specifications detailed in the literature [18], depicted in Fig 2b. Additionally, two airflow deflectors segment the outlet into four sub-outlets.

## 2.2. Heating Unit

The heating unit comprises a heater and an electric heating plate. The heat flow fan operates on the principle that ambient air is heated by the heater, followed by the fan drawing in and expelling the heated air. This electric heater warms the volute, which then introduces the heated air into the fan, facilitating heat exchange with the volute. As depicted in Fig 3, the heater operates on the principle wherein diesel, drawn from the tank by a pump, flows into the combustion chamber. An ignition plug ignites the mixture, and the resulting exhaust gases are vented through an exhaust port. The combustion chamber features an aluminum alloy exterior adorned with cooling fins for rapid heat conduction. Heated air is directed into the combustion chamber by the fan. The section equipped with fins represents the combustion chamber, while the fan is positioned to the right. For effective heating of the volute, the electric heating plate should mirror the shape of the fan volute and be appropriately affixed.

The air flow that is heated by the heater will undergo a heat exchange, enter the fan interior and take place with the volute. The heat exchange equation is as follows [19]:

$$Q_h = G \times C_P \times (T_2 - T_1)$$

(5)

Where: $Q_h$ is the heat absorption by airflow (kW); G is the air mass flow rate (kg/s); $C_P$ is the air specific heat capacity (j/(kg·°C)); $T_2$ is the heated air temperature (°C); $T_1$ is the ambient air temperature (°C).

$$Q_h = Q_{h1} + Q_e$$

(6)

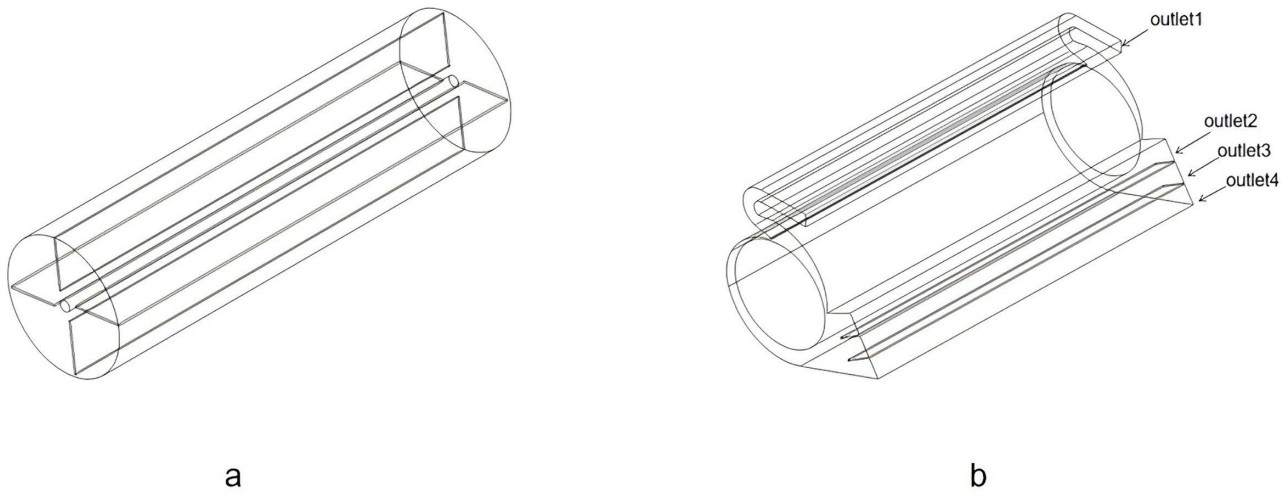

a b

**Fig 2. Structure of the centrifugal fan(a) Impeller structure, (b) Volute structure.**

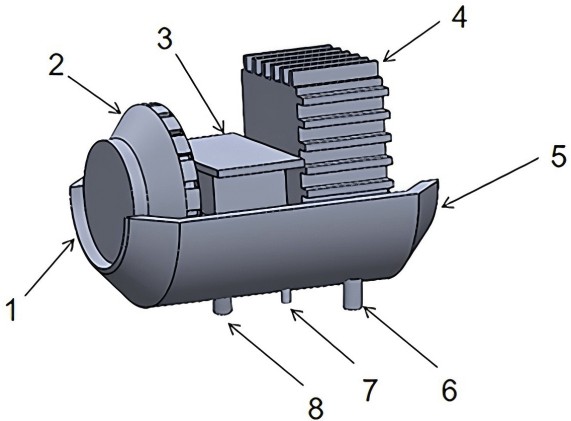

**Fig 3. Internal structure of the heater.** 1.Air Inlet 2.Fresh Air Wheel 3.Controller 4.Combustion Chamber 5.Hot Air Outlet 6.Exhaust Gas Outlet 7.Oil Inlet 8.Combustible Air Inlet.

$$Q_{h1} = \frac{3595N}{A} \tag{7}$$

$$Q_e = \frac{U^2}{R} \tag{8}$$

Where: N is the heater power (kW); A is the margin factor; $Q_{h1}$ is the heater's heat (J); U is the voltage (V); R is the resistance (Ω); $Q_e$ is the electric heater heat (J).

## 3. Numerical Model and Computational Domain

### 3.1. Numerical model

The governing equations are given in Eqs. (9) – (11) [20]:

$$\frac{\partial \rho}{\partial t} + \nabla \cdot (\rho v) = 0 \tag{9}$$

$$\rho \left( \frac{\partial v}{\partial t} + v \cdot \nabla v \right) = -\nabla p + \nabla \cdot \tau + f \tag{10}$$

$$\rho c_p \frac{\partial T}{\partial t} + \rho c_p v \cdot \nabla T = \nabla \cdot (k \nabla T) \tag{11}$$

During the functioning of the cleaning system, airflow is usually in a turbulent state and the standard k–ε turbulence model has been successfully applied in the calculation analysis of agricultural centrifugal fan. In the present study, this model is again used for the calculation of the airflow field in the cleaning system, as shown in Eqs.(12) and (13) [21]:

$$\frac{\partial}{\partial t} (\rho k) + \frac{\partial}{\partial x_i} (\rho k v_i) = \frac{\partial}{\partial x_j} \left[ \left( \mu + \frac{\mu_t}{\sigma_k} \right) \frac{\partial k}{\partial x_j} \right] + G_k - \rho \epsilon \tag{12}$$

   

$$\frac{\partial}{\partial t}\left(\rho\epsilon\right) + \frac{\partial}{\partial x_i}\left(\rho\epsilon v_i\right) = \frac{\partial}{\partial x_j}\left[\left(\mu + \frac{\mu_t}{\sigma_k}\right)\frac{\partial\epsilon}{\partial x_j}\right] + C_{1\epsilon}\frac{\epsilon}{k}G_k - C_{2\epsilon}\rho\frac{\epsilon^2}{k} + S_\epsilon \qquad (13)$$

### 3.2. Computational Domain, Mesh Generation, and Boundary Conditions

In order to investigate the effect of inlet geometry on fan performance, two different inlet shapes were modeled and compared with the original design, as seen in fig 4.

Fig 5 illustrates the computational domain of the fan. The mesh encompasses two distinct regions: the moving region and the stationary region. The moving region, which interacts with the rotating blades, is represented as a sliding mesh domain. This allows for variable solutions using the sliding mesh technique provided by ANSYS FLUENT 2021. To ensure accurate turbulence solutions proximate to the walls, denser elements were deployed within

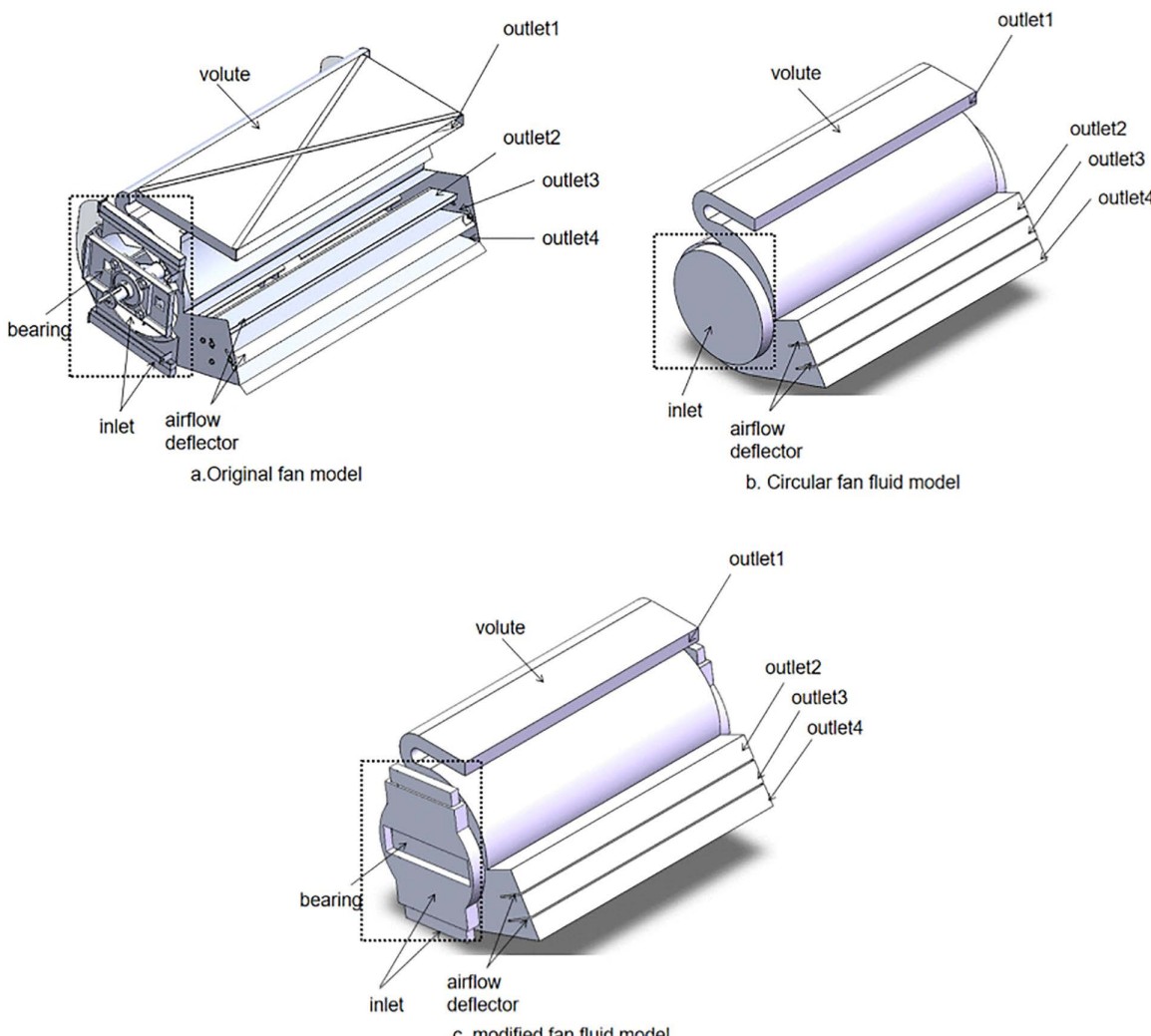

**Fig 4. Structure of the fan.**

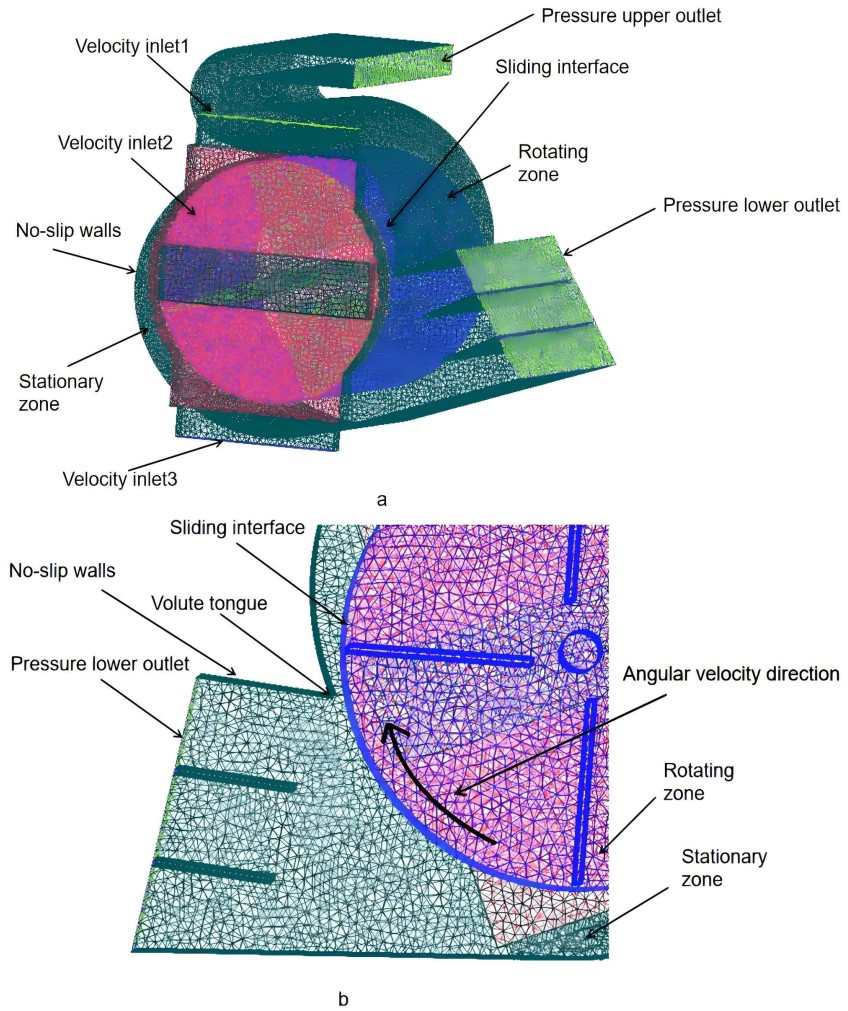

**Fig 5. Computational domain of the fan (a) Computational domain, grid system and definition of boundary conditions, (b) Detail of the mesh in the interaction region.**

the boundary layer adjacent to all walls, ensuring that the midpoint of each contiguous cell is situated within the logarithmic law domain [22].

### 3.3. Grid Independence

Grid independence, a crucial factor, was validated through simulations of the designed hot-air fan by generating three distinct grid types: coarse, medium, and fine. Four models, with cell counts spanning from 3,292,964–7,763,990, exhibited varying degrees of grid resolution. Specifically, the grid numbers were identified as 3,292,964 (grid1), 4,266,728 (grid2), 5,895,912 (grid3), and 7,763,990 (grid4). As depicted in Fig 6, at a fan speed of 800 rpm, the average airflow velocity at the outlet was assessed. It was observed that the velocity for the fine grid deviated 2.43% less than that of the coarse grid [12]. For resource conservation and enhanced computational efficiency, the simulation employed the coarse meshing strategy with a total of 3,292,964 grids.

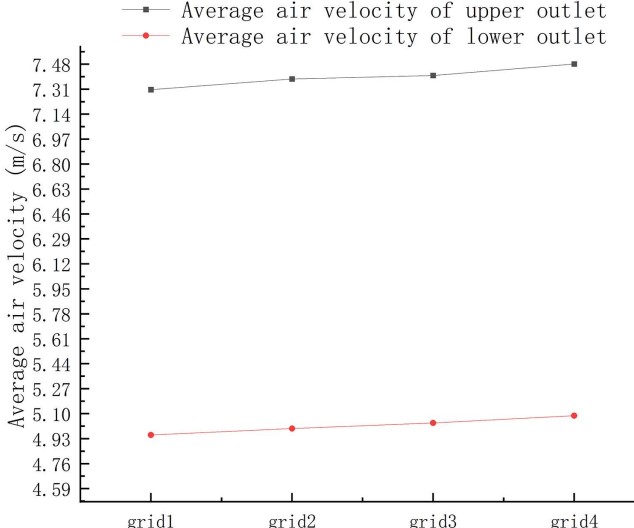

**Fig 6. Sensitivity of air velocity to grid resolution.**

## 3.4. Setting of Boundary Conditions

The inlet velocity and the outlet pressure were set as 6 m/s and 0 Pa, respectively. The diffusion term in the governing equation was discretized using a central difference scheme, while the convection term was handled with a second-order upwind scheme. The impeller wall was characterized as a rotating surface with a no-slip boundary condition. The simulation was run for 1500 iterations. The settings of turbulence mode are presented in Table 1.

## 4. Simulation results analysis

### 4.1. Effects of different inlet configurations

To determine an appropriate fan inlet, the air pressure and velocity distributions of both the circular inlet fan and the modified-inlet fan were analyzed utilizing ANSYS FLUENT 2021 and CFD-POST 2021. At a fan speed of 1200 rpm, six XY cross-sections were established to monitor the air pressure, as shown in Fig 7. Fig 8 presents the air pressure contour for

**Table 1. k-ε model setup.**

| Options | Setting Up |
|---|---|
| Model | k-epsilon (2 eqn) |
| k-epsilon Model | Realizable |
| Near-Wall Treatment | Enhance wall Treatment |
| Model Constants | |
| $C_{1\epsilon}$ | 1.44 |
| $C_{2\epsilon}$ | 1.92 |
| $C_{\mu}$ | 0.09 |
| $\sigma_K$ | 1 |
| TKE Prandtl Number | 1 |
| TDR Prandtl Number | 1.2 |

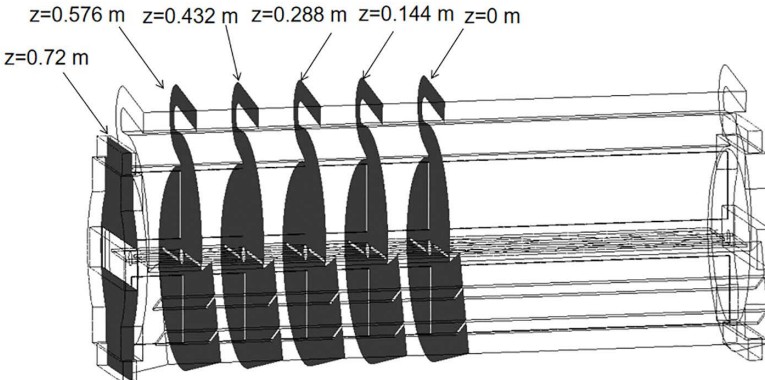

z=0.72 m   z=0.576 m   z=0.432 m   z=0.288 m   z=0.144 m   z=0 m

**Fig 7. Fan internal section.**

the two fans with different inlet configurations. It can be seen that the average air pressure variation across the XY section varies due to the distinct inlet geometries. Specifically, the air pressure for both the circular inlet fan and the modified-inlet fan demonstrates a decline within the Z = 0.576 m to Z = 0.72 m range, yet exhibits an increase from Z = 0 m to Z = 0.576 m..

Fig 9 elucidates the average air velocity of XY sections. At Z = 0.72 m, the airflow remains unaffected by the impeller acceleration, resulting in elevated air pressure. The rotation of the impeller reduces the internal air pressure, creating a pressure differential, which compels air into the fan. However, as the axial distance increases, flow velocity diminishes, reaching its nadir at Z = 0 m. It should be noted that air velocity is inversely related to air pressure; hence, proximity to the fan's central plane leads to a gradual augmentation in air pressure. The circular structure, being more uniform than the modified configuration, is less prone to vortex generation, leading to reduced air pressure in the circular air inlet fan. Evidently, the variance in air pressure, attributed to differing inlet geometries, significantly impacts fan performance.

Fig 10a,b depict the air velocity contours for the modified-inlet fan and the circular inlet fan. A marked discrepancy in the airflow distribution of the two fans can be observed at outlet 4. Notably, the air velocity of the circular inlet fan at position A exceeds that of the modified – inlet fan. Moreover, at outlet 4, the airflow distribution of the circular inlet fan is less uniform than that of the modified-inlet fan, with pronounced velocity fluctuations.

The average air velocity at the outlet of different fans was compared, as shown in Fig 11.The modified-inlet fan has an outlet average air velocity very close to the original fan of the combine harvester, whereas the circular inlet fan exhibits a higher velocity. From the above analysis, it can be concluded that the fan with the modified-inlet structure exhibits optimal air distribution, air velocity, and air pressure.

## 4.2.  Influence of Fan Speed on Air Temperature and Velocity

To investigate the influence of fan speed on air temperature and velocity, the intermediate values for inlet air temperature (80 °C) and volute temperature (40 °C) were maintained constant. Fan speed was varied to elucidate the interrelationship between fan speed, air temperature and air velocity, and results were shown in Fig 12. When the fan speed increases from 800 rpm to 1200 rpm, the upper outlet's air temperature drops from 70.25 °C to 67.09 °C, a decrement of 4.5%. Concurrently, air velocity increases from 7.31 m/s to 9.15 m/s, a rise of 25.17%. meanwhile, at the lower outlet, the air temperature decreases from 58.86 °C to 47.74 °C, while its air velocity increases from 4.96 m/s to 5.29 m/s.

Furthermore, upon elevating the speed from 1200 rpm to 1600 rpm, the upper outlet's air temperature descends from 67.09 °C to 60.671 °C, a drop of 9.57%, while its velocity surges from 9.15 m/s to 11.05 m/s, registering a 20.77% ascent. Similarly, the lower outlet witnesses its temperature drop from 47.74 °C to 43.55 °C (an 8.78% reduction), and its velocity leap from 5.29 m/s to 6.62 m/s, an increment of 25.14%.

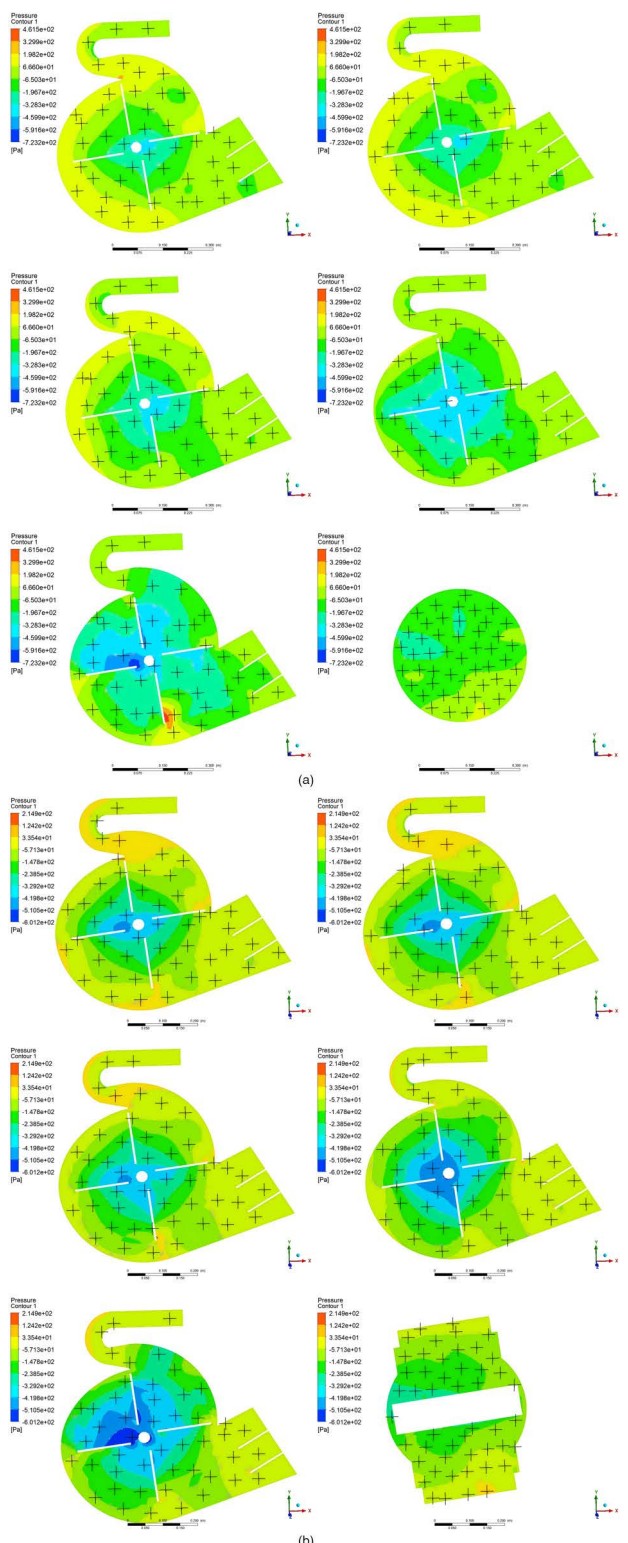

**Fig 8. Comparison of air pressure contour for XY sections.** (a) Air pressure contour for circular inlet structure; (b) Air pressure contour of the modified-inlet structure.

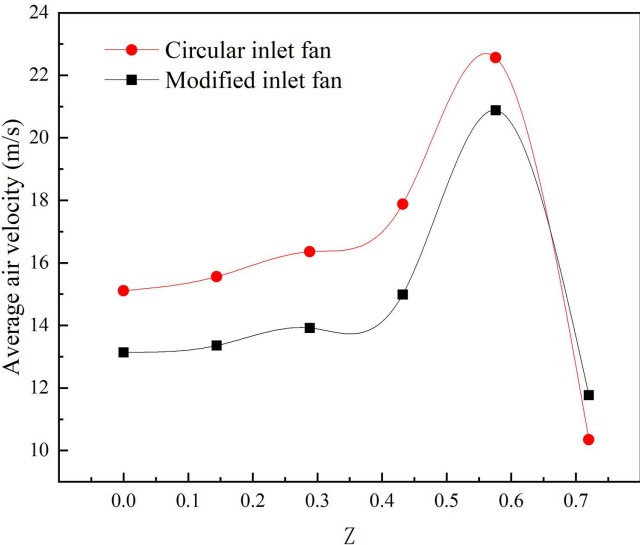

**Fig 9. Average air velocity of XY sections.**

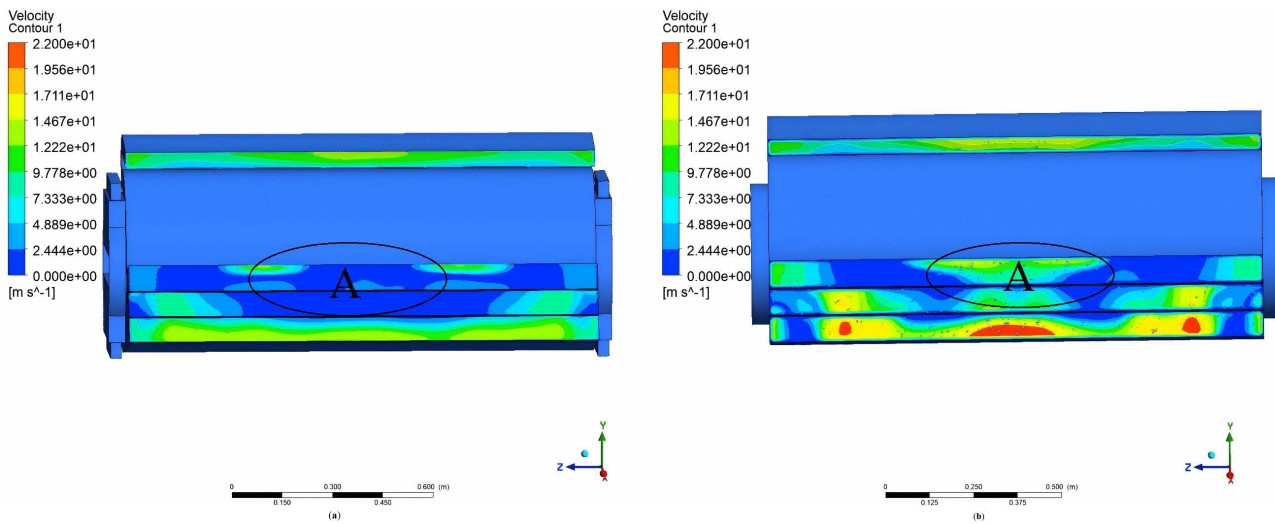

**Fig 10. Comparison of air velocity contour at the outlet.** (a) modified-inlet fan contour at the outlet; (b) Circular inlet fan contour at the outlet.

### 4.3. Influence of the Inlet Air Temperature on the Outlet Air Temperature

To investigate the influence of inlet air temperature on the fan's outlet air temperature, both the fan speed and volute temperature were held constant at 1200 rpm and 40 °C, and the inlet air temperature was 40 °C, 80 °C, and 120 °C, respectively.

Fig 13 plots the outlet air temperature versus inlet air temperature. It can be seen that the outlet air temperature rises with the increase in inlet air temperature. At an inlet temperature of 40 °C, the discrepancy in average air temperatures between the outlets remains minimal. However, as the inlet temperature rises, the temperature difference between the

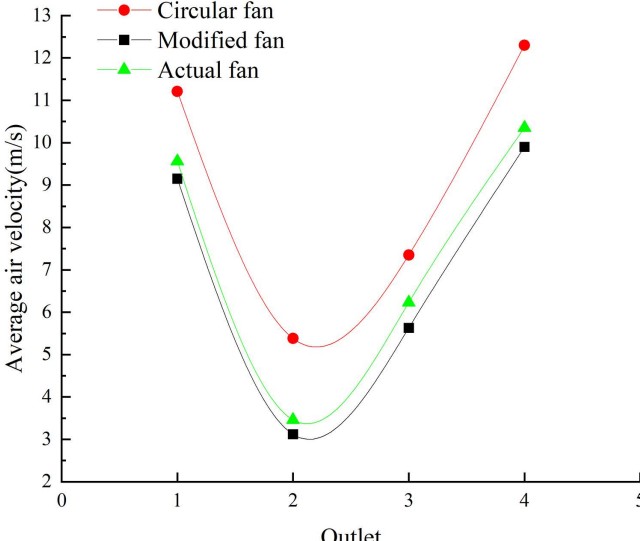

**Fig 11. Comparison of air velocity of different fans.**

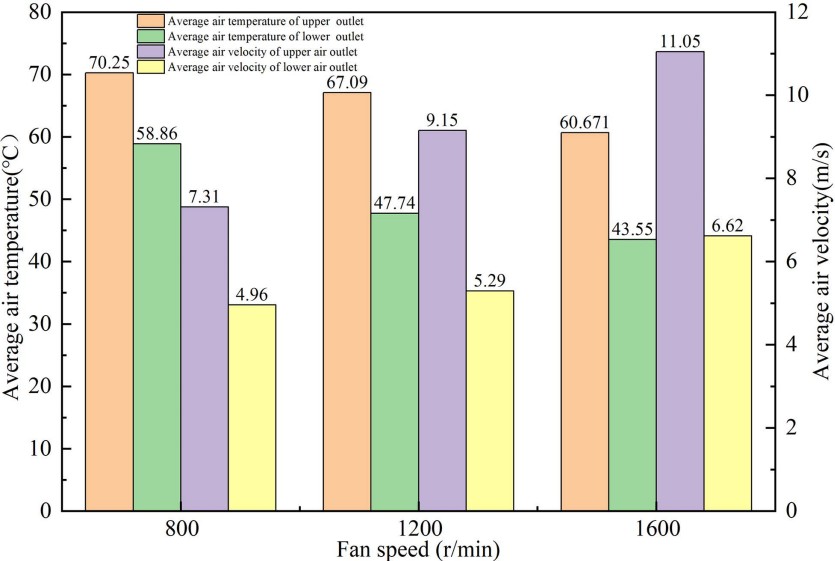

**Fig 12. Effect of fan speed on air temperature and air velocity.**

average air temperatures of the upper and lower outlets amplifies. This disparity can be attributed to the vortex disrupting the uniform airflow distribution at the lower outlet, which prevents the hot airflow from effectively reaching the outlet. Additionally, the large space of the lower outlet facilitates rapid heat dissipation, resulting in a lower average air temperature. In contrast, the upper outlet has a more uniform airflow distribution with minimal vortex interference. Coupled with its faster airflow velocity, confined space, and minimal heat loss, the upper outlet consistently shows a higher average air temperature.

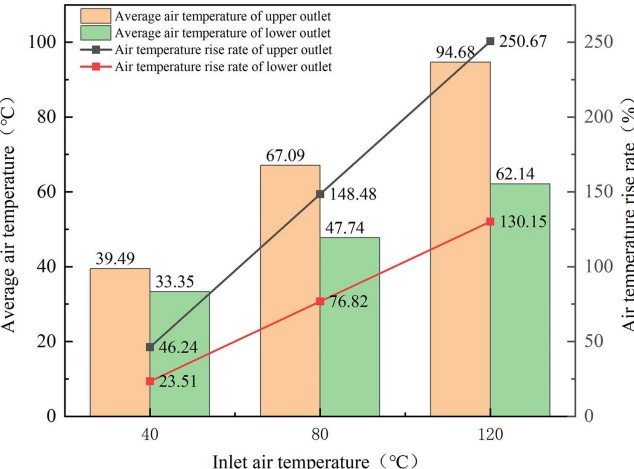

**Fig 13. Relationship between inlet air temperature and average air temperature of the fan oulet.**

### 4.4. Influence of Volute Temperature on Air Temperature

To investigate the influence of volute temperature on the fan's outlet air temperature, both the fan speed and inlet temperature were held constant at 1200 rpm and 80 °C, respectively. Fig 14 shows the air temperature contour at volute temperature of 60°C. The air temperatures are higher at outlets 1 and 4 than at outlets 2 and 3. This difference is attributed to airflow patterns: the flow is smooth at outlets 1 and 4, but significant vortices occur at outlets 2 and 3.

The dependence of the outlet air temperature on the volute temperature is shown in Fig 15, which quantifies the associated average changes. A 20 °C increase in volute temperature results in an average air temperature rise of 6 °C at the upper outlet (a 19% increase) and of 2.5 °C at the lower outlet (a 9% increase). The higher rate of temperature increase at the upper outlet is due to its constrained space and slow heat dissipation. In contrast, heat dissipation at the lower outlet is more efficient owing to ample space, a thinner thermal boundary layer, and consequently, less heat transfer resistance.

### 4.5. Response surface tests

The Box–Behnken design approach [2 4] was undertaken to assess the interaction of fan speed, inlet air temperature, and volute temperature on the fan's outlet air velocity and temperature. The test results are presented in Table 2.

Based on the data of Table 2, Design-Expert 13.0 software was employed to derive the quadratic polynomial regression model, as shown in equation(27)-equation(30).

$$Y_1=68.24-5.64A+27.10B+3.93C+0.3875AB0.0425AC\ 0.9075BC-2.22A^2\ 0.0597B^2\ 0.6997C^2 \tag{14}$$

$$Y_2=47.65-7.45A+14.96B+2.40C+0.0837AB-0.0005AC+0.0035BC+3.45A^2-0.0492B^2+0.0135C^2 \tag{15}$$

$$Y_3=9.15+1.84A+0.0239A^2+0.0004B^2-0.0001C^2 \tag{16}$$

$$Y_4=5.28+0.8050A+0.0001B-0.0001C-0.0005AB-0.0002BC+0.4909A^2+0.0002B^2-0.0003C^2 \tag{17}$$

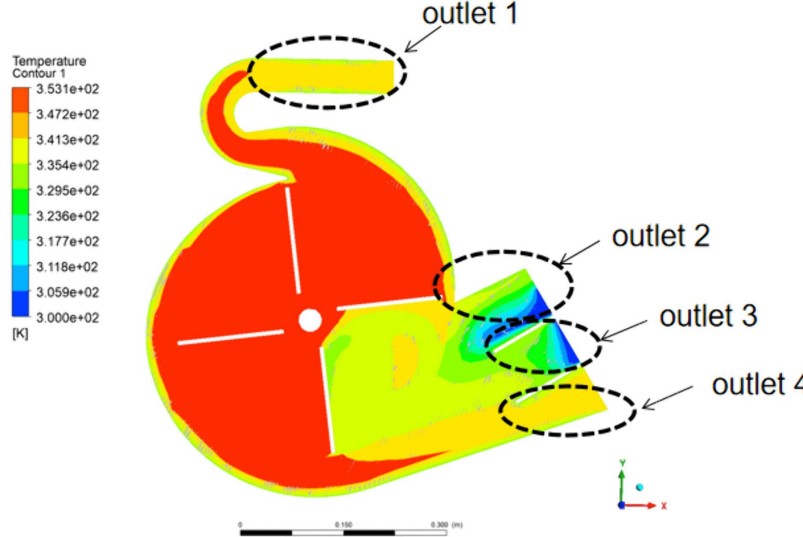

**Fig 14. Air temperature contour at volute temperature of 60 °C and z = 0.324 m.**

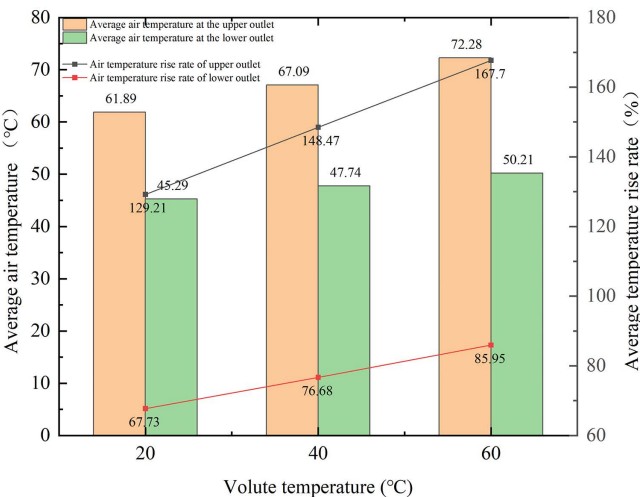

**Fig 15. Relationship between volute temperature and average air temperature of outlet.**

In the model, $Y_1$, $Y_2$, $Y_3$ and $Y_4$ represent the average air temperature at the upper outlet, signifies the average air temperature at the lower outlet, the average air velocity at the upper outlet and the average air velocity at the lower outlet, respectively. Furthermore, A, B and C indicates the fan speed, the air temperature at the inlet and the volute temperature, respectively. The outlet air temperature is most sensitive to inlet air temperature, then fan speed, and least to volute temperature. Conversely, the outlet air velocity is strongly governed by fan speed but largely insensitive to the two temperature variables.

3D response surface map of the interaction effects of various factors was generated based on regression model analysis results are shown in Fig 16–19. From Figs 16 and 17, the fan speed is inversely proportional to the upper outlet

**Table 2. Test results of response surface of Hot-air fan cleaning performance.**

| | Fan Speed (rpm) | Inlet Air Temperature (°C) | Volute Temperature (°C) | Average Air Temperature at the Upper Outlet (°C) | Average Air Temperature at the Lower Outlet (°C) | Average Air Velocity at the Upper Outlet (m/s) | Average Air Velocity at the Lower Outlet (m/s) |
|---|---|---|---|---|---|---|---|
| 1 | 1600 | 80 | 20 | 55.65 | 41.218 | 11.013 | 6.571 |
| 2 | 1600 | 80 | 60 | 63.52 | 46.018 | 11.013 | 6.571 |
| 3 | 1200 | 80 | 40 | 68.02 | 47.618 | 9.146 | 5.275 |
| 4 | 800 | 80 | 60 | 75.08 | 61.018 | 7.327 | 4.961 |
| 5 | 800 | 80 | 20 | 67.04 | 56.216 | 7.327 | 4.961 |
| 6 | 1200 | 80 | 40 | 68.13 | 47.723 | 9.146 | 5.275 |
| 7 | 1200 | 80 | 40 | 68.09 | 47.689 | 9.147 | 5.275 |
| 8 | 1600 | 120 | 40 | 88.15 | 58.659 | 11.014 | 6.571 |
| 9 | 1200 | 80 | 40 | 68.09 | 47.615 | 9.146 | 5.276 |
| 10 | 1600 | 40 | 40 | 32.69 | 28.636 | 11.013 | 6.572 |
| 11 | 1200 | 40 | 60 | 43.59 | 35.015 | 9.147 | 5.275 |
| 12 | 1200 | 120 | 20 | 89.56 | 60.215 | 9.146 | 5.276 |
| 13 | 800 | 40 | 40 | 44.55 | 43.618 | 7.327 | 4.961 |
| 14 | 1200 | 40 | 20 | 37.66 | 30.218 | 9.147 | 5.275 |
| 15 | 800 | 120 | 40 | 98.46 | 73.306 | 7.328 | 4.962 |
| 16 | 1200 | 80 | 40 | 68.88 | 47.626 | 9.146 | 5.276 |
| 17 | 1200 | 120 | 60 | 99.12 | 65.026 | 9.146 | 5.275 |

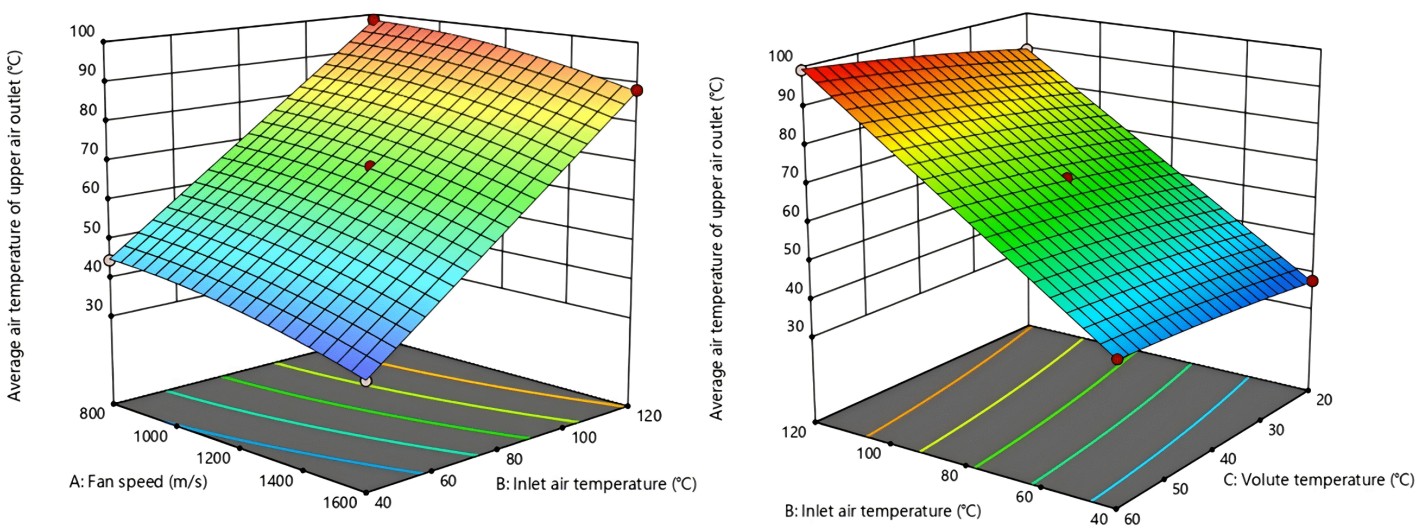

**Fig 16. Response surface of the air temperature interaction effect at the upper outlet.**

temperature and lower outlet temperature, while the volute surface temperature is directly proportional to that. Nonetheless, the interaction degree is greater for inlet air temperature and fan speed than for inlet air temperature and volute temperature. This indicates that inlet air temperature, compared to fan speed and volute surface temperature, has a more dominant influence on the outlet air temperature.

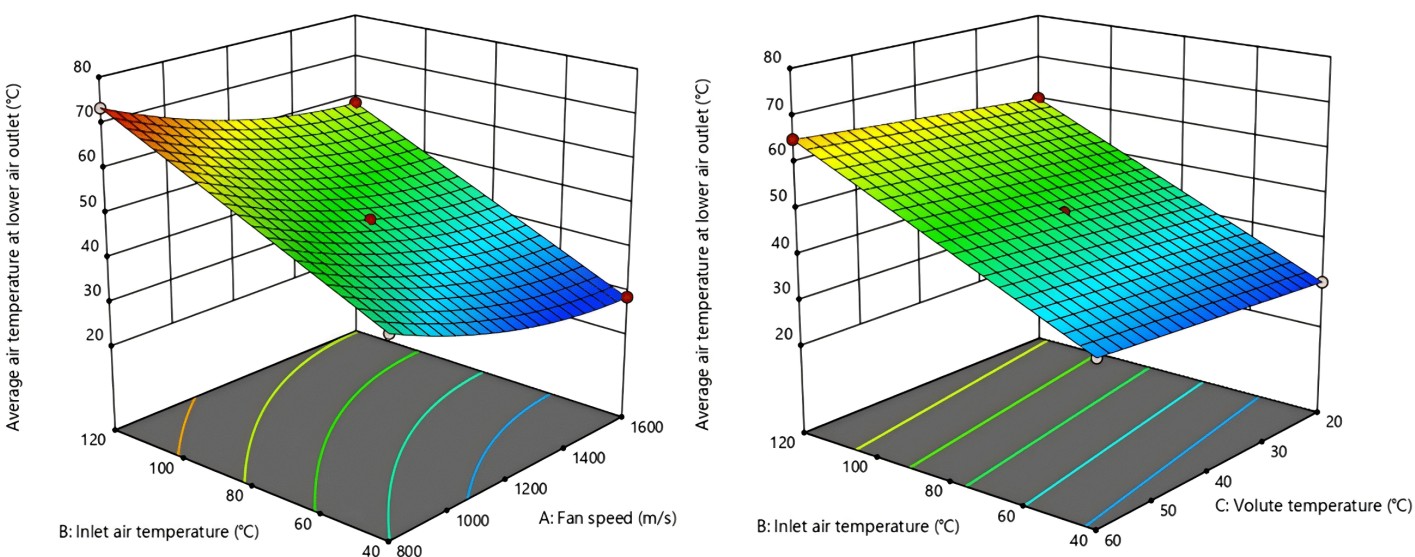

**Fig 17. Response surface of the air temperature interaction effect at the lower outlet.**

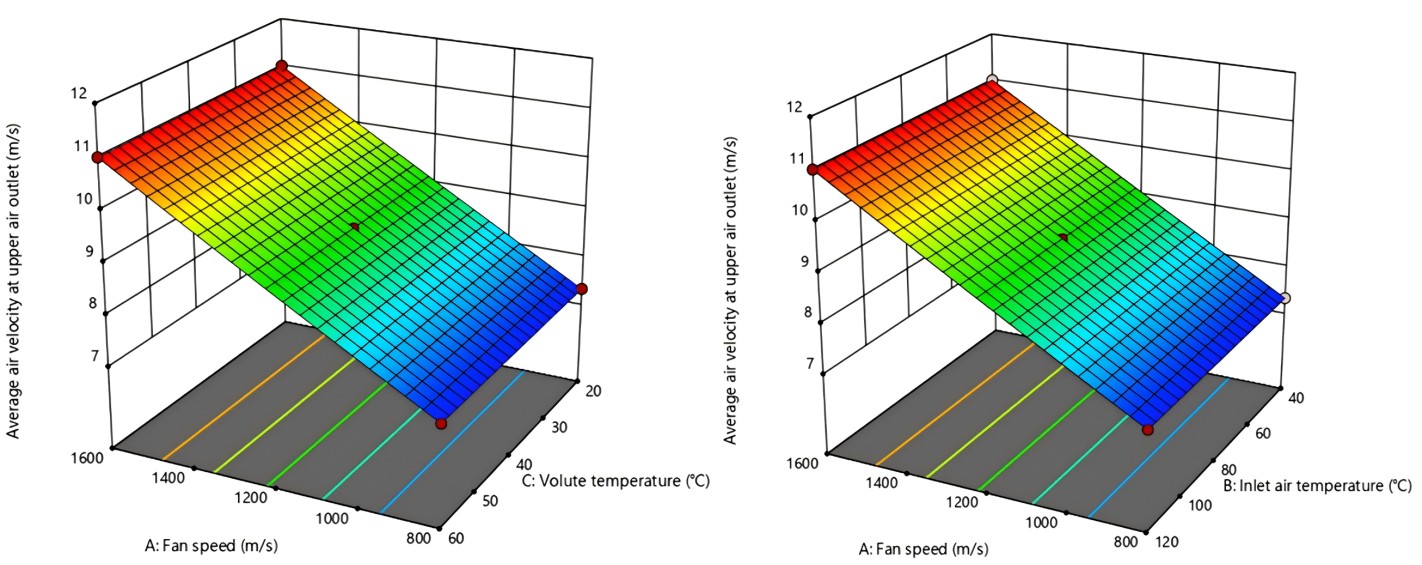

**Fig 18. Response surface for the air velocity interaction effect at the upper outlet.**

From Figs 18 and 19, air velocity of the upper and lower outlet is unaffected by changes in inlet air temperature and volute temperature. In contrast, when these temperatures are held constant, variations in fan speed cause significant changes in air velocity of upper and lower outlet. This indicates that the outlet air velocity is predominantly influenced by fan speed

Based on the optimization module of Design-Expert, the optimal parameter combination was obtained as:a fan speed of 1445 rpm, inlet air temperature of 85 °C, and volute temperature of 60 °C. Under the optimal parameter combination,

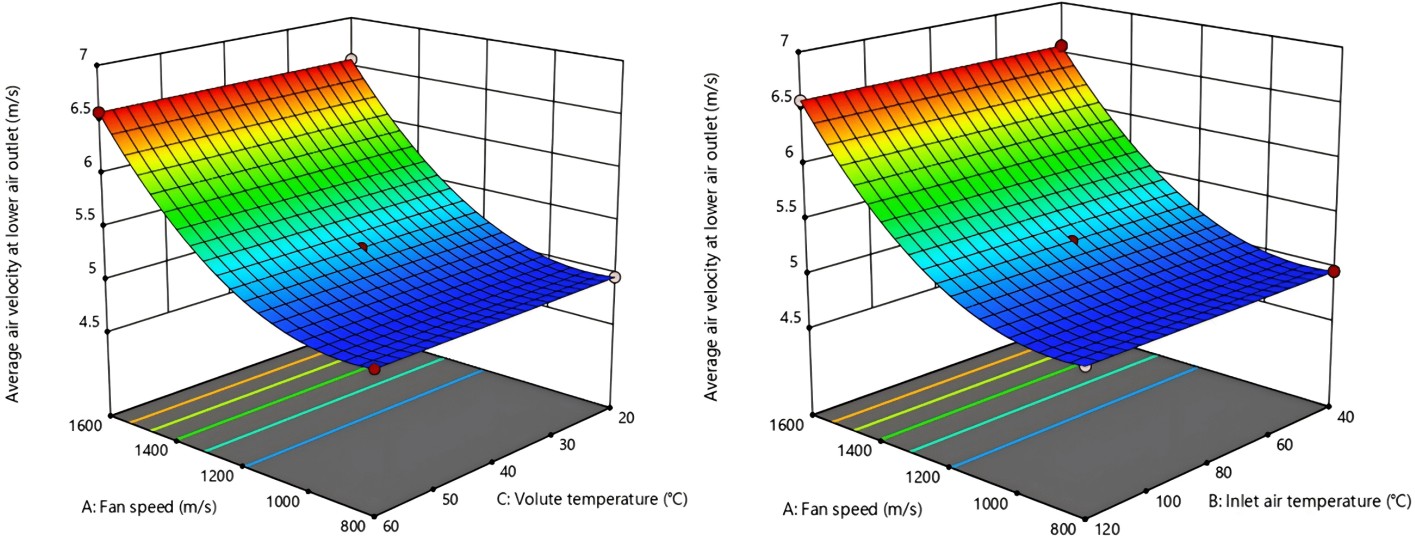

**Fig 19. Response surface of the air velocity interaction effect at the lower outlet.**

the air temperatures at the upper and lower outlets were 67 °C and 46 °C, respectively. Correspondingly, the air velocities were 10.3 m/s and 6 m/s.

## 5. Bench test

The bench test was conducted to validate the simulation outcomes and the bench structure is shown in Fig 20. The heater was positioned on each side of the fan with a rated power of 20 kW and a flow rate of 1.63 m³/s. Under the optimal parameter combination, averages air velocity and temperature of oulet 1,2,3 and 4 were measured to corroborate the simulation findings.

Fig 21 depicted the measured air velocity and temperature at each fan outlet. The wind speed at Outlet 1 is the highest, while that at Outlet 3 is the lowest. Outlet 4 exhibits the smallest flow fluctuation and the highest transverse uniformity. Conversely, Outlet 3 has the largest flow fluctuation and the least uniform flow distribution. Outlet 1 exhibits the smallest temperature fluctuation and the most uniform temperature distribution. In contrast, Outlet 3 has the largest fluctuation and a significantly less uniform distribution.

The average air velocity and temperature of air outlet 1 are 10.6 m/s and 65 °C, respectively. The average air velocity and temperature of air outlets 2, 3, and 4, which represent the average value of the lower outlets, were calculated to be 6.9 m/s and 51 °C, respectively. Therefore, the average air velocity and temperature are basically consistent with the theoretical optimal value.

## 6. Discussion

A comparison between the numerical simulation and the experimental test shows that the errors at the upper outlet are 2.8% for air velocity and 3.1% for air temperature. In contrast, the errors at the lower outlet are significantly larger: 13% for air velocity and 9.8% for air temperature. The error can be attributed to several factors. Firstly, pressure pulsations generated by the rotation of the four-bladed fan cause fluctuations in air velocity. Secondly, discrepancies exist between the boundary conditions used in the simulations and the actual conditions [23,24]. This implies that numerical simulations cannot entirely substitute for experimental measurements in informing fan structural design. Nonetheless, numerical simulations remain a viable tool for investigating the fan's internal airflow organization.

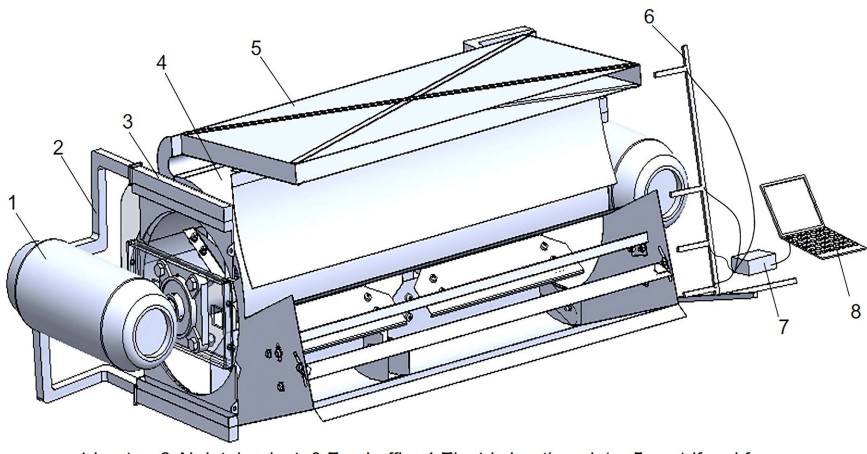

1.heater  2.Air intake duct  3.Fan baffle  4.Electric heating plate  5.centrifugal fan
6.hot-wire anemometer  7.anemometer bracket  8.computer

**Fig 20.  Experimental setup components of the hot-air fans.**

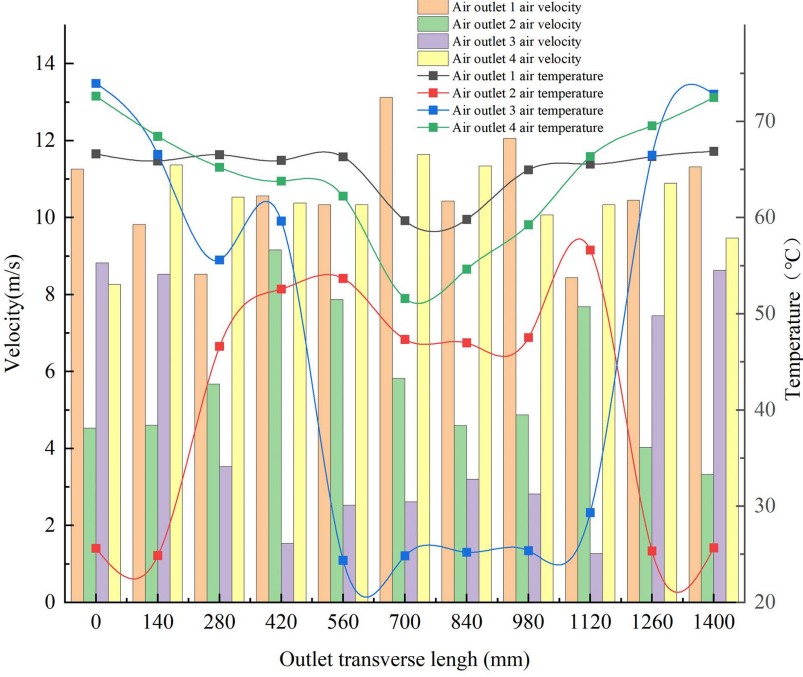

**Fig 21.  Transverse distribution of airspeed and air temperature for hot-air fans.**

Hot-air experimentation discerned that a separation effect of moist crops is optimized when the air velocity is 7.6 m/s, and the wind temperature is 60 °C [25]. In this study, the airflow velocity of designed hot-air cleaning fan ranges from 3.5 to 10.4 m/s, and the airflow temperature ranges from 25.7 to 66.3 °C. Hence, the airflow velocity and temperature of designed hot-air cleaning fan were able to seperate moist crops. Future studies will concentrate on the behavior of wet particles within the cleaning chamber influenced by the hot air flow.

## 7. Conclusions

In an endeavor to enhance the cleaning efficiency of the combine harvester, this study introduced a novel hot-air cleaning fan..Based on the aforementioned research outcomes, the following primary conclusions were derived:

(1) By comparing the simulation and actual performance differences between the circular and modified-inlet fans, it was determined that the modified-inlet structure could more accurately simulate the actual airflow characteristics of the fan of the combine harvester and provide a reliable model basis for subsequent optimization.

(2) The outlet temperature of the fan was most significantly affected by the inlet temperature, and less affected by the volute temperature. When the speed was elevated from 800 rpm to 1200 rpm, there was a 4.5% decrease in the air temperature and a 25.17% increase in air velocity at the upper outlet. In contrast, the lower outlet exhibited an 18.9% temperature decrease and a 6.65% increase in air velocity. Upon further acceleration from 1200 rpm to 1600 rpm, the upper outlet experienced a 9.57% temperature decrease and a 20.77% velocity increase, whereas the lower outlet recorded an 8.78% decrease in temperature and a 25.14% boost in velocity..

(3) The ideal operating parameter combination of the heating fan was determined using the response surface approach, which is 1445 rpm, 85 °C for the input temperature, and 60 °C for the volute temperature. The optimization was confirmed by bench tests, which produced an outlet velocity of 4.63–10.57 m/s and a temperature of 41.16–64.96 °C, both of which satisfied the requirements for wet crop separation.

## Supporting information

**S1 Table. The "Minmal data set" of Fig 6.**
(PDF)

**S2 Table. The "Minmal data set" of Fig 9.**
(PDF)

**S3 Table. The "Minmal data set" of Fig 11.**
(PDF)

**S4 Table. The "Minmal data set" of Fig 21.**
(PDF)

## Author contributions

**Conceptualization:** guoliang you.

**Data curation:** guoliang you.

**Formal analysis:** tao zhang.

**Funding acquisition:** yaoming li.

**Investigation:** tao zhang, guoliang you.

**Methodology:** tao zhang, guoliang you.

**Resources:** tao zhang.

**Supervision:** yaoming li.

**Writing – original draft:** tao zhang.

**Writing – review & editing:** tao zhang.

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
