## [Decision Letter · Decision Letter 0]

2 Sep 2025

Dear Dr. zhang,

Thank you for submitting your manuscript to PLOS ONE. After careful consideration, we feel that it has merit but does not fully meet PLOS ONE’s publication criteria as it currently stands. Therefore, we invite you to submit a revised version of the manuscript that addresses the points raised during the review process.

We look forward to receiving your revised manuscript.

Kind regards,

SATHISH SIVASUBARAMANIYAN

Academic Editor

PLOS ONE

Journal Requirements:

“This research was funded by National Natural Science Foundation of China under Grant grant number (51975257).”

“The authors gratefully acknowledge the National Natural Science Foundation of China.”

“This research was funded by National Natural Science Foundation of China under Grant grant number (51975257).”

5. We note that your Data Availability Statement is currently as follows: All relevant data are within the manuscript and its Supporting Information files.

6. PLOS requires an ORCID iD for the corresponding author in Editorial Manager on papers submitted after December 6th, 2016. Please ensure that you have an ORCID iD and that it is validated in Editorial Manager. To do this, go to ‘Update my Information’ (in the upper left-hand corner of the main menu), and click on the Fetch/Validate link next to the ORCID field. This will take you to the ORCID site and allow you to create a new iD or authenticate a pre-existing iD in Editorial Manager.

Reviewers' comments:

Reviewer's Responses to Questions

**Comments to the Author**

1. Is the manuscript technically sound, and do the data support the conclusions?

Reviewer #1: Yes

2. Has the statistical analysis been performed appropriately and rigorously?

Reviewer #1: No

3. Have the authors made all data underlying the findings in their manuscript fully available?

Reviewer #1: Yes

4. Is the manuscript presented in an intelligible fashion and written in standard English?

Reviewer #1: Yes

Reviewer #1: 1. Literature review is not up to date.

2. Need for the present study is not clearly spelt out.

3. Governing equations invoked is not specified

4. Grid independence study not available

5. Turbulence models study not done

6. Detail discussion on the findings between the improvement observed in the current and modified models not clearly specified

7. Conclusion to be rewritten with focus on the current findings

**Do you want your identity to be public for this peer review?** For information about this choice, including consent withdrawal, please see our Privacy Policy

Reviewer #1: No

---

## [Author Response · Author response to Decision Letter 1]

30 Oct 2025

respond to editor:

1.The manuscript has been revised according to PLOS ONE's style requirements.

2.The sponsor is responsible for overseeing the grammar and formatting of the paper during the study.

3.I have removed the funding statement from my paper and would like to update it through the online submission form. The updated statement will be“Funding: This research was funded by National Natural Science Foundation of China under Grant grant number (51975257). ”

5.Data Availability Statement:All relevant data are within the manuscript.

6. I have an ORCID iD and that it has been validated in Editorial Manager.

7.The reviewer comments do not include a recommendation to cite specific previously published works.

respond to reviewer:

Comment no.1�：Literature review is not up to date.

Response�：As for the problem of outdated references you mentioned, we have carefully checked and updated them. The newly added references will better support our research arguments and enable our study to be more effectively connected with and compared to previous works. Lines 85-95

Comment no.2�：Need for the present study is not clearly spelt out.

Response�：Thank you for your valuable advice,the need for the present study has been added in the article, see lines 120-127.

Comment no.3�：Governing equations invoked is not specified

Response�：Thank you for your valuable advice,The governing equations invoked have been specified. Line 216�，eqs 9-11.

Comment no.4�：Grid independence study not available

Response�：Thank you for your valuable advice, the grid independence study is presented in section 3.3, line 234.

Comment no.5�：Turbulence models study not done

Response�：Thank you for your valuable advice,Turbulence model of k-epsilon was added from Eqs12 and 13,see line 220.

Comment no.6�：Detail discussion on the findings between the improvement observed in the current and modified models not clearly specified

Response�：Thank you for your valuable advice,Detail discussion on the findings between the improvement observed in the current and modified models has been added in the Sec 9�，line637-639. The details are“In the present investigation, discrepancies between the model and the actual fan yield variances between simulation and test results. As shown in Fig 12 and Fig 23, the air velocity of the circular inlet fan and actual fan maximum discrepancies 18%. The maximum discrepancies of air velocity between the modified fan and the actual fan is 8.7%. The discrepancy in the error between the two types of fans arises from the model's structure.By comparing the simulation results with the test results, although the air velocity of the outlet of the improved fan is smaller than that of the circular inlet fan, it is closer to the air velocity of the actual fan. The simulated experimental conditions tend to be more idealistic than the actual experimental conditions.”

Comment no.7�：Conclusion to be rewritten with focus on the current findings

Response�：Thank you for your valuable advice,the Conclusion has been rewritten.The details are“In an endeavor to enhance the cleaning efficiency of the combine harvester, this study introduced a novel hot-air cleaning fan. Through an integration of FLUENT 2021 simulation analysis and experimental results from Design Expert 13.0, it was determined that the modified hot-air cleaning fan demonstrates superior performance. Based on the aforementioned research outcomes, the following primary conclusions were derived:

(1)By comparing the simulation and actual performance differences between the round and improved inlet fans, it was determined that the improved inlet structure could more accurately simulate the actual airflow characteristics of the fan and provide a reliable model basis for subsequent optimization.

(2)The outlet temperature of the fan was most significantly affected by the inlet temperature, and less affected by the volute temperature. The outlet wind speed was mainly determined by the rotation speed, and was positively correlated.

(3)The ideal operating parameter combination of the heating fan was determined using the response surface approach, which is 1445 rpm, 85 °C for the input temperature, and 60 °C for the volute temperature. The optimization was confirmed by bench tests, which produced an outlet velocity of 4.63–10.57 m/s and a temperature of 41.16–64.96 °C, both of which satisfied the requirements for wet crop separation. ”

---

## [Decision Letter · Decision Letter 1]

18 Dec 2025

Dear Dr. zhang,

Thank you for submitting your manuscript to PLOS ONE. After consideration of the reviewer reports and my own assessment, I find that the study shows potential but **does not yet meet PLOS ONE’s**
**publication**
**criteria** in its current form. I am therefore inviting you to submit a **revised version** of the manuscript.

The reviewers’ comments are provided below. While Reviewer #3 considers that many issues raised previously have been addressed, both reviewers identify **remaining concerns related to manuscript structure, clarity of presentation, and completeness of reporting** that must be resolved before the manuscript can be considered further.

Please note the following points, which should be addressed in your revision:

The manuscript requires **substantial revision for standard academic English** , including grammar and sentence structure. As PLOS ONE does not provide copyediting the revised manuscript must be clear and publication-ready.The overall **structure of the manuscript should be revised****,** including consolidation of multiple “results and discussion” and “conclusion” sections into single, appropriately organized sections.All equations taken or adapted from previous literature (including Equations 1–5) must be **properly cited****,** and this should be ensured consistently throughout the manuscript.The **logical relationship between CFD simulation, bench testing, and response surface methodology** should be clearly explained and justified.The presentation of **verification results** should be clarified, and overlapping sections should be merged where appropriate.The **conclusion section should be revised to be more concise and to emphasize quantitative findings****.**Table 3 should be reformatted to comply with PLOS ONE table guidelines.The **literature review should be strengthened** , including the incorporation of more recent relevant studies.

When revising your manuscript, please provide a **detailed, point-by-point response** to each reviewer comment, indicating how the issues have been addressed or explaining your reasons if you have chosen not to follow a specific suggestion.

We look forward to receiving your revised manuscript.

Kind regards,

**Soheil Mohtaram**

Academic Editor

PLOS ONE

Journal Requirements:

Reviewers' comments:

Reviewer's Responses to Questions

**Comments to the Author**

Reviewer #2: (No Response)

Reviewer #3: All comments have been addressed

2. Is the manuscript technically sound, and do the data support the conclusions?

Reviewer #2: No

Reviewer #3: Yes

3. Has the statistical analysis been performed appropriately and rigorously?

Reviewer #2: Yes

Reviewer #3: N/A

4. Have the authors made all data underlying the findings in their manuscript fully available?

Reviewer #2: Yes

Reviewer #3: Yes

5. Is the manuscript presented in an intelligible fashion and written in standard English?

Reviewer #2: No

Reviewer #3: No

Reviewer #2: Below are my comments regarding the manuscript:

- The manuscript does not adhere to standard English structures. A thorough revision is necessary to edit the structures.

- The article is lengthy, and this volume is somewhat too extensive for publication in the journal. Please condense the content to make it more concise and effective.

- Please provide the references for each of the equations (1 to 5) that you have taken from other sources. Additionally, please ensure this for all other equations as well.

- Why have terms such as "results and discussion" and "conclusion" been used multiple times in different sections, like 3.3, 5, 9, and 10? This format is not suitable for publication in a reputable journal like PLOS ONE. Every article should have a single conclusion, so please review and revise the entire paper from the beginning.

- The format of Table 3 is not suitable for publishing in this journal.

- After the overall edits, you can resubmit it for review.

Reviewer #3: The author has basically solved the existing problems according to the reviewer's comments, but there are still some obvious shortcomings in the manuscript.

1. The structure of the introduction needs to be modified, there is no need to divide it into so many paragraphs. In addition, the content in the last paragraph, from line 102 to line 106, starting with "in conclusion", is unnecessary and not suitable for writing in the introduction.

2. I agree with the previous reviewer's opinion that the literature review is still very insufficient. Currently, the cited literature is too old, mostly published before 2015. Please review the literature published in the past five years and add it to the manuscript. Otherwise, the current literature review cannot reflect the necessity and innovation of your research.

3. Section 4.2, where is the results of the verification ? Is it included in Section 6? If so, the two should be merged.

4. Please the authors to explain the logical relationship between CFD simulation, bench testing, and response surface methodology analysis ? Why is it necessary to use all three methods simultaneously in one research ?

5. The conclusion is still too long and lacks quantitative results.

**Do you want your identity to be public for this peer review?** For information about this choice, including consent withdrawal, please see our Privacy Policy

Reviewer #2: No

Reviewer #3: No

---

## [Author Response · Author response to Decision Letter 2]

24 Feb 2026

Response to editor:

1.The manuscript requires substantial revision for standard academic English, including grammar and sentence structure.

Response�：The grammar and sentence structure of this manuscript have been revised .

2.The overall structure of the manuscript should be revised, including consolidation of multiple “results and discussion” and “conclusion” sections into single, appropriately organized sections。

Response�：The multiple “results and discussion” and “conclusion”sections have been consolidated into single.

3.All equations taken or adapted from previous literature (including Equations 1–5) must be properly cited, and this should be ensured consistently throughout the manuscript.

Response�：All equations taken or adapted from previous literature have been properly cited.

4.The logical relationship between CFD simulation, bench testing, and response surface methodology should be clearly explained and justified.

Response�：The logical relationship between CFD simulation, bench testing, and response surface methodology has been clearly explained and justified.

5.The presentation of verification results should be clarified, and overlapping sections should be merged where appropriate.

Response�：The presentation of verification results has been clarified, and overlapping sections have been merged where appropriate.

6.The conclusion section should be revised to be more concise and to emphasize quantitative findings.

Response�：The conclusion section has been revised to be more concise and the quantitative findings have been emphasized.

7.Table 3 should be reformatted to comply with PLOS ONE table guidelines.

Response�：The format of Table 3 has been revised.

8.The literature review should be strengthened, including the incorporation of more recent relevant studies.

Response�：The literature review has been revised and more recent relevant studies have been cited.

Response to reviewer2:

Comment no.1�：The manuscript does not adhere to standard English structures. A thorough revision is necessary to edit the structures.

Response�：Thank you for your valuable advice, the grammar and sentence structure of this manuscript have been revised .

Comment no.2�：The article is lengthy, and this volume is somewhat too extensive for publication in the journal. Please condense the content to make it more concise and effective.

Response�：Thank you for your valuable advice, the content of this article has been condensed.

Comment no.3�：Please provide the references for each of the equations (1 to 5) that you have taken from other sources. Additionally, please ensure this for all other equations as well.

Response�：Thank you for your valuable advice,all the references for each of the equations that taken from other sources have been provided, such as equations 9-13,see line 121 and 125.

Comment no.4�：Why have terms such as "results and discussion" and "conclusion" been used multiple times in different sections, like 3.3, 5, 9, and 10? This format is not suitable for publication in a reputable journal like PLOS ONE. Every article should have a single conclusion, so please review and revise the entire paper from the beginning。

Response�：Thank you for your valuable advice, the multiple “results and discussion” and “conclusion”sections have been consolidated into single.

Comment no.5�：The format of Table 3 is not suitable for publishing in this journal

Response�：Thank you for your valuable advice, the format of Table 3 has been revised, see line 256.

Response to reviewer3:

Comment no.1�：The structure of the introduction needs to be modified, there is no need to divide it into so many paragraphs. In addition, the content in the last paragraph, from line 102 to line 106, starting with "in conclusion", is unnecessary and not suitable for writing in the introduction.

Response�：Thank you for your valuable advice,the structure of the introduction has been condensed and the content in the last paragraph has been revised.

Comment no.2�：I agree with the previous reviewer's opinion that the literature review is still very insufficient. Currently, the cited literature is too old, mostly published before 2015. Please review the literature published in the past five years and add it to the manuscript. Otherwise, the current literature review cannot reflect the necessity and innovation of your research。

Response�：Thank you for your valuable advice,the literature review has been revised and more recent relevant studies have been cited.

Comment no.3�：Section 4.2, where is the results of the verification ? Is it included in Section 6? If so, the two should be merged.

Response�：Thank you for your valuable advice, the results of the verification is included in section 6 and the two have be merged in section 5.

Comment no.4�：Please the authors to explain the logical relationship between CFD simulation, bench testing, and response surface methodology analysis ? Why is it necessary to use all three methods simultaneously in one research ?

Response�：Thank you for your valuable advice. Firstly, the simulation provides a theoretical basis for the design of the inlet structure of the hot airflow fan. Then, the working parameters are optimized through the response surface method. Finally, the simulation results were verified through the bench test. All three methods were used together in this study, forming a coherent chain from theoretical analysis to experimental validation.

Comment no.5�：The conclusion is still too long and lacks quantitative results.

Response�：Thank you for your valuable advice,the conclusion section has been revised to be more concise and the quantitative findings have been emphasized.

---

## [Editor Report · Decision Letter 2]

25 Feb 2026

**Manuscript Number:** PONE-D-25-43359R2

**Title:** Numerical Simulation of a Hot-Air Cleaning Fan for the Combine Harvester

Dear Dr. Zhang,

We are pleased to inform you that your manuscript entitled “Numerical Simulation of a Hot-Air Cleaning Fan for the Combine Harvester” has been **accepted for publication in PLOS ONE** , pending final processing checks by the journal office.

The manuscript will now proceed to the production stage, where it will undergo technical checks and preparation for publication. You may be contacted by the production team if any additional information or clarification is required.

Thank you for choosing PLOS ONE for the dissemination of your research. We appreciate your contribution and look forward to seeing your work published.

Congratulations on your acceptance.

Sincerely,

Soheil Mohtaram

Academic Editor

PLOS ONE

---

## [Editor Report · Acceptance letter]

PONE-D-25-43359R2

PLOS One

Dear Dr. zhang,

I'm pleased to inform you that your manuscript has been deemed suitable for publication in PLOS One. Congratulations! Your manuscript is now being handed over to our production team.

Kind regards,

on behalf of

Dr. Soheil Mohtaram

Academic Editor

PLOS One